# STI testing and subsequent clinic attendance amongst test negative asymptomatic users of an internet STI testing service; one-year retrospective study

**Oluseyi Ayinde[1]\*, Louise Jackson[2], Jara Phattey[1], Jonathan D. C. Ross[1]**

1 Sexual Health and HIV, University Hospitals Birmingham NHS Trust, Birmingham, United Kingdom,
2 Health Economics Unit, Institute of Applied Health Research, College of Medical and Dental Sciences, University of Birmingham, Birmingham, United Kingdom

\* oluseyi.ayinde@uhb.nhs.uk

**Data Availability Statement:** All relevant data of the study are anonymized and published within the paper. The routine data that support the findings of this study are available from the University

## Abstract

### Aim

To explore the characteristics of online STI test users, and assess the frequency and factors associated with subsequent service use following a negative online STI test screen in individuals without symptoms.

### Methods

One-year retrospective study of online and clinic STI testing within a large integrated sexual health service (Umbrella in Birmingham and Solihull, England) between January and December 2017. A multivariable analysis of sociodemographic and behavioural characteristics of patients was conducted. Sexual health clinic appointments occurring within 90 days of a negative STI test, in asymptomatic individuals who tested either online or in clinic were determined. Factors associated with online STI testing and subsequent clinic use were determined using generalized estimating equations and reported as odds ratios (OR) with corresponding 95% confidence intervals (CI).

### Results

31 847 online STI test requests and 40 059 clinic attendances incorporating STI testing were included. 79% (25020/31846) of online STI test users and 49% (19672/40059) of clinic STI test takers were asymptomatic. Online STI testing was less utilised (p<0.05) by men who have sex with men (MSM), non-Caucasians and those living in neighborhoods of greater deprivation. Subsequent clinic appointments within 90 days of an asymptomatic negative STI test occurred in 6.2% (484/7769) of the online testing group and 33% (4960/15238) for the clinic tested group. Re-attendance following online testing was associated with being MSM (aOR 2.55[1.58 to 4.09]—MSM vs Female) and a recent prior history of STI testing (aOR 5.65[4.30 to 7.43] 'clinic tested' vs 'No' recent testing history).

Hospitals Birmingham NHS Health Informatics, but restrictions apply to the availability of these data, which were used under ethics approval for the current study, and so are not publicly available. Data are however available from the authors upon reasonable request depending on an individual permission from the UHB Research Governance Office (R&D@uhb.nhs.uk).

**Funding:** The author(s) received no specific funding for this work.

**Competing interests:** JDCR reports personal fees from Bayer and GSK Pharma, as well as ownership of shares in GSK Pharma and AstraZeneca Pharma; and is author of the UK and European Guidelines on Pelvic Inflammatory Disease; is a Member of the European Sexually Transmitted Infections Guidelines Editorial Board. He is an NIHR Journals Library Editor and associate editor of Sexually Transmitted Infections journal. He is an officer of the International Union against Sexually Transmitted Infections (treasurer), and a charity trustee of the Sexually Transmitted Infections Research Foundation (chair). OA, LJ, and JP declare no competing interests or funding. This does not alter our adherence to PLOS ONE policies on sharing data and materials.

## Conclusions

Subsequent clinic attendance amongst online STI test service users with negative test results was infrequent, suggesting that their needs were being met without placing an additional burden on clinic based services. However, unequal use of online services by different patient groups suggests that optimised messaging and the development of online services in partnership with users are required to improve uptake.

## Introduction

Over the last decade, sexual health services have adopted innovative ways to increase testing and screening for sexually transmitted infections (STI) in response to a reduction in available resources [1]. Home-based STI self-sampling test kits available via web-based 'online' and pharmacy-based platforms provide an alternative to traditional face to face clinic services and can facilitate an increase in STI testing by detecting asymptomatic infections that would have otherwise remained undiagnosed [2–4] and reaching groups who have not previously been tested [4–7]. Online STI self-sampling kits have comparable diagnostic accuracy to samples collected by clinicians [8, 9], can reduce the time to test [5], relieve pressure on traditional clinics [10], and can be cost-effective when integrated with other sexual health services [11]. Also, from a user perspective home-based STI self-sampling can increase autonomy and control of the testing process, and reduce the self-perceived stigma of attending a sexual health clinic [12–14].

However, the benefits of online STI testing may be attenuated by a number of factors including limited uptake in high-risk groups, poor return rates of test kits, inequity in health or digital literacy of potential users, and increasing the pressure in an overstretched clinical service due to increased testing and follow up [9, 11, 15, 16]. Studies on online STI testing in the United Kingdom, suggest that a preference for online STI testing was associated with being female and Caucasian [2, 3, 6, 17]. Up to 60% of online STI kits requested are not returned [6, 11, 15] and non-return of kits was commoner in younger individuals, ethnic minorities (Black African/Caribbean or mixed-race), men, and those who were symptomatic at the time of testing [6, 15, 17]. Overall, this highlights the potential for reduced uptake of online based STI testing within certain high risk groups such as in men who have sex with men (MSM) and Black African/Caribbean ethnic groups.

In addition, assessing the value of online testing is dependent on a full understanding of the patient pathway including service utilisation after online self-sampling, for example, if patients' needs are not met fully via the online service then they will re-attend the clinical service within a short period with an associated increase in costs and service pressure, and reduced satisfaction. To assess this requires longer term follow up to determine the frequency of reuse of sexual health services following online based STI testing, and evaluation of the factors associated with re-attendance. This is of particular relevance given the rapid expansion of online based STI testing following the COVID-19 pandemic. However, there is paucity of data on patient outcomes in the time period after online self-sampling. Therefore the objectives of this study were (i) to explore the characteristics of users of web-based and clinic based STI testing over a one-year time period, (ii) to assess the frequency of clinic attendance within 90 days following an online test request amongst asymptomatic online STI test users who tested negative, and (iii) to identify factors associated with subsequent sexual clinic attendance in this group.

## Methods

### Study design and data sources

Umbrella health offers free sexual health services for residents in Birmingham and Solihull, England. A retrospective cohort study of Umbrella STI testing service use was conducted as part of the 'Understanding the costs and benefits of online screening and clinic-based screening for sexually transmitted infections (STIs): a case study of the Umbrella sexual health system in Birmingham' project. Data on STI testing, sociodemographic, behavioural and symptom characteristics of patients who were tested for STIs at the sexual health clinic were extracted from the electronic patient record (EPR) (Excelicare®, AxSys Technology Ltd). The corresponding data for those requesting online STI testing were obtained from the Umbrella STI self-sampling kit request website (https://umbrellahealth.co.uk/self-sampling-kits/), the website associated database, and the EPR. On return of an online test kit, STI kit order details were transferred from the website database into the EPR with matching of the web database and EPR patient identification numbers. Data on test results and subsequent clinic attendance after baseline STI testing were then retrieved from the EPR.

### Study inclusion and exclusion criteria

Participants were included if they were aged 16 years or over, and requested STI testing (chlamydia, gonorrhea, syphilis, HIV or hepatitis B) through the Umbrella online self-testing internet website or attended the sexual health clinic for STI testing between 1st of January 2017 and 31st of December 2017. For the online test cohort, participants were excluded if the requested test kits could not be matched to the participant's online request identification number. In the clinic cohort, those attending clinic solely for aspects of sexual healthcare that could not be provided on the online testing website such as clinic attendance for contraceptives or advice were excluded.

### Outcomes

**Online STI testing completed.**   This was defined as the request and return of a postal online STI self-sampling kit from the Umbrella STI testing internet website within the one-year study duration.

**Clinic use following an asymptomatic online STI test.**   This was defined as having booked an appointment to attend a sexual health clinic within 90 days of requesting an online STI test in an individual who:

i.  was asymptomatic and tested negative for all tests

ii.  had received their test results

Clinic appointments booked after 31st of December 2017 were excluded. When multiple online test requests were booked within the 90-day period, only the latest request was included. The 90-day follow up period was chosen on the assumption that test users were likely to reuse sexual health services within a short period if their needs from an initial engagement were unmet, and to limit the risk of incorporating routine appointments as the guidelines at the time recommended that high risk STI groups retest 3-monthly [18].

### Statistical analysis

Descriptive analyses were performed on demographic, behavioral and clinical data. Univariable and multivariable associations were estimated with generalized estimating equations to

account for repeat use of the same testing service (online or clinic) during the study period. Odds ratios (ORs) and the corresponding 95% confidence intervals (CI) were used to estimate associations with the outcome variables (i) online STI testing completed, and (ii) clinic use following an asymptomatic online STI test. Due to the collinearity between the gender and sexuality variables, these variables were combined into a composite 'gender /sexuality' variable. To limit the risk of unmeasured confounding all variables including an interaction term for hepatitis B and gender /sexuality were included in the multivariable model. This was because MSM who test negative for hepatitis B were advised to attend clinic for hepatitis B vaccination. Statistical analyses on the utilization of online STI testing were stratified by symptom presentation at the time of test request, as symptomatic patients using the Umbrella sexual health service were encouraged to attend clinic for testing, and because data from England suggest that most (>75%) online STI test takers are asymptomatic [2–4]. No imputation was performed for missing data and case wise deletion for missing data was used in the regression models. Two-tailed tests were used and p values<0.05 were considered statistically significant. Statistical analyses were conducted using STATA 16 for Windows.

## Ethical review and considerations

A 'no objection to study' was obtained from the University Hospitals Birmingham NHS Trust for the 'Understanding the costs and benefits of online screening and clinic-based screening for sexually transmitted infections (STIs): a case study of the Umbrella sexual health system in Birmingham' with project reference number RRK6031. Ethical review and approval for the 'Understanding the costs and benefits of online screening and clinic-based screening for sexually transmitted infections (STIs): a case study of the Umbrella sexual health system in Birmingham' project was also provided by the University of Birmingham's Science, Technology, Engineering and Mathematics Ethical Review Committee with ethical review reference number–ERN_17–1065. As this was an analysis of routinely collected health information, patient consent was not required.

## Results

Between January 1st 2017 and December 31st 2017, 164 015 clinic STI test records (gonorrhea, chlamydia, HIV, syphilis or hepatitis B) and 44 044 clinic attendances were identified. Of these, 3758 clinic attendances were excluded because there was no record of an STI test relating to these attendances. 227 clinic attendances were also excluded because test users were less than 16 years old. Therefore 40 059 clinic STI test attendances (33 812 of these were unique attendances and the rest from repeat attendances during the study period) were included in the analysis.

Similarly, 76 776 online STI test records (gonorrhea, chlamydia, HIV, syphilis or hepatitis B) were linked to 31 901 online STI test user requests. Of these, 54 online user requests could not be matched to an STI kit being sent out and were excluded. Therefore, 31 847 online user requests for STI test kits were included in the analysis. Of these, 31 846 had records of the symptom status at the time of test request. About half of the online STI test kits requested and sent out (14539/31847 [46%]) were not returned for processing. Of the 17 308 kits returned for processing 14 655 were unique kit requests and the rest were repeat requests within the study period.

19672/40059 (49%) of the clinic STI test attendances were from asymptomatic individuals and 25020/31846 (79%) of the online STI test request were from asymptomatic individuals.

## Online STI testing preference

In Table 1, the socio-demographic and test characteristics of service users are presented with odds ratios comparing online and clinic patients who were tested for STIs, stratified by the presence or absence of symptoms. Amongst asymptomatic patients the odds for undertaking online STI testing were significantly lower amongst older adults (adjusted odds ratio—aOR 0.82[0.73 to 0.93] aged over 40 years vs 16 to 20 year), men (aOR 0.55[0.52 to 0.59] male vs female) or MSM (aOR 0.13[0.12 to 0.16] MSM vs female), those self-reporting non-Caucasian ethnicity (aOR 0.28 [0.25 to 0.31]—Asian vs Caucasian; aOR 0.25[0.23 to 0.27]—Black vs Caucasian), contacts of STIs (aOR 0.66[0.61 to 0.71]) and those with previous attendance at the clinic within the last three months (aOR 0.90 [0.84 to 0.96]). The odds for having had an online STI test was higher in those living in neighborhoods with less deprivation (aOR1.41 [1.32 to 1.51]; aOR 1.23 [1.13 to 1.34]—for Index of Multiple Deprivation [IMD] quintile 2 and 3 respectively compared to IMD quintile 1), and for being tested for gonorrhea and/or chlamydia (aOR 2.07 [1.74 to 2.47]).

Amongst symptomatic individuals who were tested, the odds for online STI testing were broadly similar to asymptomatic individuals, except that the odds for online STI testing was significantly higher for contacts of an STI (aOR 1.36[1.22 to 1.51]) Table 1.

## Subsequent clinic use within 90 days of a negative online STI test in asymptomatic individuals

17308/31847 (54%) STI test kits were returned following an online test request, of which 3436/17308 [20%] individuals were symptomatic at the time of testing and 1303/17308 (8%) tested positive for at least one STI. 3249/17308 (19%) tests were 'indeterminate' or 'not tested' and in 1551/17308 (10%) there was no record of the patients having successfully received a notification of their test results–these patients were excluded from the analysis on subsequent clinic attendance. Overall, 7769/17308 (45%) online test kits returned and tested were from asymptomatic individuals who tested negative and had received test results.

The frequency of booking a clinic appointment within 90 days of having had an online asymptomatic STI screen and testing negative was 6.2% (484/7769—Table 2), with a median time to clinic booking of 34 days (interquartile range[IQR], 18 to 58 days), and with most appointments leading to a clinic attendance 410/484 (85%). The factors associated with a clinic appointment following a negative asymptomatic STI screen are presented in Table 3 and were significantly higher in those self-identifying as MSM (aOR 2.55 [1.58 to 4.09]—MSM vs Female), previously testing for STIs in the three months prior to the online test (aOR 5.65 [4.30 to 7.43] previous clinic STI testing vs 'No' prior STI testing; aOR 3.12[2.42 to 4.03]—previous home-based STI kit testing vs 'No' prior STI testing), and for online requested self-sample test kits collected from the clinic (aOR 1.98 [1.46 to 2.68]) or pharmacy (aOR 1.50 [1.03 to 2.20]) compared to test kits delivered by post to home addresses. Conversely, the odds of a subsequent clinic appointment were significantly lower in those living in neighborhoods with less deprivation (aOR 0.77 [0.60 to 0.98]; 0.69 [0.50 to 0.95]; 0.58 [0.36 to 0.91] for IMD quintile 2, 3 and 4 respectively compared to IMD quintile 1) and completing an online test for HIV and/or syphilis (aOR 0.74 [0.59 to 0.92]).

To help interpret the rate of clinic use following a negative online STI test, we also explored subsequent booking of a clinic appointment in those who had initially been tested in clinic. Of the 40 059 clinic attendees, 15 238 were asymptomatic at the time of testing and tested negative. For this group, the frequency of clinic appointment booking in the 90 days after attending clinic for asymptomatic STI tests and testing negative was 33% (4960/15238), with a median time to subsequent clinic appointment of 29 days (IQR- 14 to 57 days). 3191/4960 (64%) of

**Table 1. STI testing modality and associated factors (n = 71906) .**

| | | Asymptomatic (n = 44692/71905[62.2]) | | | | | | Symptomatic (n = 27213/71905[37.8]) | | | | | |
|---|---|---|---|---|---|---|---|---|---|---|---|---|---|
| | | Clinic STI tested (n = 19672/44692[44.0]) | Online STI tested (n = 13872/44692 [31.0]) | Non-return of online test kits (n = 11148/44692[24.9]) | online vs clinic STI tested | | | Clinic STI tested (n = 20387/27213[74.9]) | Online STI tested (n = 3436/27213[12.6]) | Non-return of online test kits (n = 3390/27213[12.5]) | online vs clinic STI tested | | |
| | | | | | OR (95% CI) | aOR (95% CI) | P value | | | | OR (95% CI) | aOR (95% CI) | P value |
| Age (median[IQR]), in years Min;Max | | 26(21 to 33) 16;84 | 24 (21 to 29) 16;76 | 24 (20 to 29) 16;75 | NA | NA | NA | 26 (22 to 23) 16;90 | 24 (21 to 29) 16;75 | 23 (20 to 28) 16;83 | NA | NA | NA |
| Age, n(%)* | 16 to 20 | 4025 (20.5) | 2925 (21.1) | 2858 (25.7) | 1 | 1 | <0.001 | 3846 (18.9) | 744 (21.7) | 917 (27.1) | 1 | 1 | <0.001 |
| | 21 to 30 | 9548 (48.5) | 8221(59.3) | 6135 (55.1) | 1.15 (1.08 to 1.22) | **1.50 (1.40 to 1.61)** | | 9887 (48.5) | 2044 (59.5) | 1824 (53.8) | 1.06 (0.96 to 1.16) | **1.20 (1.08 to 1.33)** | |
| | 31 to 40 | 3805 (19.3) | 1916 (13.8) | 1504 (13.5) | 0.68 (0.63 to 0.74) | **1.20 (1.09 to 1.31)** | | 4163 (20.4) | 473 (13.8) | 472 (13.9) | 0.59 (0.52 to 0.67) | **0.82 (0.71 to 0.94)** | |
| | over 40 | 2294 (11.7) | 810 (5.8) | 645 (5.8) | 0.47 (0.43 to 0.52) | **0.82 (0.73 to 0.93)** | | 2491 (12.2) | 175 (5.1) | 175 (5.2) | 0.37 (0.31 to 0.44) | **0.53 (0.44 to 0.64)** | |
| Gender/ Sexuality, n(%) ‡ | Female | 9422 (48.0) | 9306 (67.1) | 6881 (61.9) | 1 | 1 | <0.001 | 11723 (57.5) | 2371 (69.0) | 2161(64.1) | 1 | 1 | <0.001 |
| | Male | 7664 (39.0) | 3055 (22.0) | 3432 (30.9) | 0.42 (0.40 to 0.44) | **0.55 (0.52 to 0.59)** | | 7311 (35.9) | 774 (22.5) | 926 (27.5) | 0.52 (0.47 to 0.57) | **0.67 (0.61 to 0.74)** | |
| | MSM | 2546 (13.0) | 1496 (10.8) | 803(7.2) | 0.63 (0.58 to 0.68) | **0.13 (0.12 to 0.16)** | | 1339 (6.6) | 289 (8.4) | 283 (8.4) | 1.10 (0.95 to 1.26) | **0.23 (0.18 to 0.31)** | |
| Ethnicity, n (%)* | Asian | 1954 (9.9) | 610 (4.4) | 887 (8.0) | 0.25 (0.22 to 0.27) | **0.28 (0.25 to 0.31)** | <0.001 | 2503 (12.3) | 196 (5.7) | 335 (9.9) | 0.24 (0.21 to 0.28) | **0.29 (0.25 to 0.35)** | <0.001 |
| | Black | 4246 (21.6) | 1199 (8.6) | 1309 (11.7) | 0.22 (0.21 to 0.24) | **0.25 (0.23 to 0.27)** | | 4278 (21.0) | 273 (7.9) | 334 (9.9) | 0.21 (0.18 to 0.24) | **0.23 (0.20 to 0.27)** | |
| | Not stated | 3434 (17.5) | 252 (1.8) | 375 (3.4) | 0.03 (0.03 to 0.04) | **0.05 (0.04 to 0.06)** | | 3397 (16.7) | 56 (1.6) | 115 (3.4) | 0.03 (0.02 to 0.04) | **0.05 (0.04 to 0.07)** | |
| | Other | 2071 (10.5) | 1401 (10.1) | 1141 (10.1) | 0.50 (0.46 to 0.54) | **0.49 (0.44 to 0.53)** | | 2390 (11.7) | 407 (11.9) | 391 (11.5) | 0.52 (0.47 to 0.59) | **0.53 (0.46 to 0.60)** | |
| | White | 7967 (40.5) | 10410 (75.0) | 7436 (66.7) | 1 | 1 | | 7819 (38.4) | 2504 (72.9) | 2213 (65.3) | 1 | 1 | |
| Index of Multiple Deprivation Quintile, n(%)* | 20% (most deprived) | 9624 (49.9) | 5224 (37.8) | 4525 (40.8) | 1 | 1 | <0.001 | 11001 (54.9) | 1625 (47.5) | 1687 (50.1) | 1 | 1 | <0.001 |
| | 20% to 40% | 4438 (23.0) | 4276 (30.9) | 3426 (30.9) | 1.87 (1.77 to 1.98) | **1.41 (1.32 to 1.51)** | | 4247 (21.2) | 890 (26.0) | 816 (24.2) | 1.47 (1.34 to 1.61) | 1.09 (0.98 to 1.20) | |
| | 40% to 60% | 2375 (12.3) | 2309 (16.7) | 1660 (15.0) | 1.80 (1.67 to 1.93) | **1.23 (1.13 to 1.34)** | | 2342 (11.7) | 465 (13.6) | 424 (12.6) | 1.39 (1.24 to 1.56) | 0.90 (0.80 to 1.02) | |
| | 60% to 80% | 1427 (7.4) | 1165 (8.4) | 859 (7.8) | 1.50 (1.37 to 1.64) | 0.95 (0.86 to 1.06) | | 1298 (6.5) | 279 (8.2) | 243 (7.2) | 1.40 (1.21 to 1.62) | 0.87 (0.74 to 1.01) | |
| | 80 to 100% (least deprived) | 1436 (7.4) | 850 (6.2) | 609 (5.5) | 1.05 (0.95 to 1.15) | **0.59 (0.52 to 0.66)** | | 1146 (5.7) | 164 (4.8) | 196 (5.8) | 0.99 (0.83 to 1.18) | **0.52 (0.43 to 0.62)** | |

(*Continued*)

**Table 1.** (Continued)

| | | Asymptomatic (n = 44692/71905[62.2]) | | | | | | Symptomatic (n = 27213/71905[37.8]) | | | | | |
|---|---|---|---|---|---|---|---|---|---|---|---|---|---|
| | | Clinic STI tested (n = 19672/44692[44.0]) | Online STI tested (n = 13872/44692 [31.0]) | Non-return of online test kits (n = 11148/44692[24.9]) | online vs clinic STI tested | | | Clinic STI tested (n = 20387/27213[74.9]) | Online STI tested (n = 3436/27213[12.6]) | Non-return of online test kits (n = 3390/27213[12.5]) | online vs clinic STI tested | | |
| | | | | | OR (95% CI) | aOR (95% CI) | P value | | | | OR (95% CI) | aOR (95% CI) | P value |
| Contact of STI, n(%) | No | 16017 (81.4) | 12576 (90.7) | 10158 (91.1) | 1 | 1 | <0.001 | 16831 (82.6) | 2825 (82.2) | 2784 (82.1) | 1 | 1 | <0.001 |
| | Yes | 2808 (14.3) | 1279 (9.2) | 979 (8.8) | 0.64 (0.60 to 0.68) | **0.66 (0.61 to 0.71)** | | 2737 (13.4) | 609 (17.7) | 603 (17.8) | 1.07 (0.97 to 1.17) | **1.36 (1.22 to 1.51)** | |
| | Unknown | 847 (4.3) | 17 (0.1) | 11 (0.1) | 0.10 (0.08 to 0.12) | **0.06 (0.05 to 0.09)** | | 819 (4.0) | 2 (0.1) | 3 (0.1) | 0.07 (0.04 to 0.12) | **0.05 (0.02 to 0.12)** | |
| Clinic attendance in the last three months, n(%)* | No | 16761 (85.2) | 11518 (83.1) | - | 1 | 1 | 0.002 | 17652 (86.6) | 2997 (87.5) | - | 1 | 1 | <0.001 |
| | Yes | 2911 (14.8) | 2344 (17) | - | 0.88 (0.85 to 0.93) | **0.90 (0.84 to 0.96)** | | 2735 (13.4) | 430 (12.6) | - | 0.77 (0.71 to 0.83) | **0.72 (0.64 to 0.80)** | |
| Tested for gonorrhoea/ chlamydia, n (%) | No | 692 (3.5) | 218 (1.6) | - | 1 | 1 | <0.001 | 678 (3.3) | 61 (1.8) | - | 1 | 1 | 0.001 |
| | Yes | 18980 (96.5) | 13654 (98.4) | - | 2.97 (2.59 to 3.40) | **2.07 (1.74 to 2.47)** | | 19709 (96.7) | 3375 (98.2) | - | 2.25 (1.75 to 2.88) | **1.86 (1.38 to 2.51)** | |
| Tested for HIV/ syphilis, n(%) | No | 2005 (10.2) | 3263 (23.5) | - | 1 | 1 | <0.001 | 2747 (13.5) | 846 (24.6) | - | 1 | 1 | <0.001 |
| | Yes | 17667 (89.8) | 10609 (76.5) | - | 0.44 (0.41 to 0.46) | **0.53 (0.49 to 0.57)** | | 17640 (86.5) | 2590 (75.4) | - | 0.52 (0.48 to 0.56) | **0.72 (0.65 to 0.79)** | |
| Tested for hepatitis B, n (%) | No | 13484 (68.5) | 12592 (90.8) | - | 1 | 1 | <0.001 | 14152 (69.4) | 3199 (93.1) | - | 1 | 1 | <0.001 |
| | Yes | 6188 (31.5) | 1280 (9.2) | - | 0.30 (0.28 to 0.32) | **0.01 (0.01 to 0.02)** | | 6235 (30.6) | 237 (6.9) | - | 0.19 (0.17 to 0.21) | **0.01 (0.00 to 0.02)** | |

Female includes the following sexualities; bisexual females (n = 174), women who have sex with women (n = 148), and other female (n = 9)

Male includes the following sexualities; bisexual males (n = 53) and other male (n = 6)

Indication for testing was not recorded for online test kits requested but not returned.

n = 71905 due to n = 1 missing data on symptom status

‡ non-binary gender n = 86 and 36 for asymptomatic and symptomatic STI test cohort were exclude due to the small proportion of this group

*—Does not add up to column total due to missing records

aOR- adjusted for age, gender/sexuality, ethnicity, IMD Quintile, contact of STI, clinic attendance in the last three months, gonorrhoea/chlamydia test, HIV/syphilis test, hepatitis B test and an interaction term for gender/sexuality and hepatitis B. n = 33063/33544 observations for asymptomatic testing and n = 23432/23823 observations for symptomatic testing

IQR- interquartile range; Min- minimum; Max- maximum

NA-Not assessed

Blank cells under the 'Non-return of online test kits' column are due to missing data from STI test kits that were not returned for processing

Bold fonts highlight statistically significant associations (P<0.05)

clinic bookings resulted in clinic attendance—Table 4. The odds of booking a further clinic appointment within 90 days of having a negative STI screen in clinic was significantly higher in MSM (aOR 3.41 [2.97 to 3.92]), ethnic minorities—Asian (aOR 1.19 [1.05 to 1.35]), Black (aOR 1.15 [1.04 to 1.27]), and those attending clinic for testing within the preceding three months of testing (aOR 1.10[1.02 to 1.18]). The odds for booking a further clinic appointment were significantly lower with increasing age (aOR 0.80 [0.73 to 0.88]; aOR 0.80 [0.71 to 0.90]); 0.82 [0.71 to 0.94] for age intervals 21 to 30; 31 to 40, and over 40s compared to 16 to 20 years), living in

**Table 2. Frequency of booking a clinic appointment within 90 days of a negative STI test obtained via online testing or clinic attendance.**

| Clinic appointment booked within 90 days of an asymptomatic negative online STI test (n = 484/7769 [6.2%]) | | | Clinic appointment within 90 days of an asymptomatic negative clinic STI test (n = 4960/15238[32.6%]) | | |
|---|---|---|---|---|---|
| Mean(SD), days (n = 484) | | 39 (24.5) | Mean(SD), days (n = 4960) | | 36 (24.7) |
| Median, (IQR), days (n = 484) | | 34 (18 to 58) | Median, (IQR), days (n = 4960) | | 29 (14 to 57) |
| Min;Max, days (n = 484) | | 4;90 | Min;Max, days (n = 4960) | | 1;90 |
| Frequency, n(%) | within 30 days | 220 (45.5) | Frequency, n(%) | within 30 days | 2666 (53.8) |
| | 31 to 60 days | 150 (30.9) | | 31 to 60 days | 1170 (23.6) |
| | 61 to 90 days | 114 (23.6) | | 61 to 90 days | 1124 (22.7) |
| Outcome of booked appointment (n = 484) | | | Outcome of booked appointment (n = 4960) | | |
| Attended, n(%) | | 413 (85.3) | Attended, n(%) | | 3241 (65.3) |
| Did not attend, n(%) | | 71 (14.7) | Did not attend, n(%) | | 1719 (34.7) |

SD- standard deviation; IQR- interquartile range; Min- minimum; Max- maximum

neighbourhoods with less deprivation (aOR 0.89 [0.82 to 0.98]; aOR 0.88[0.78 to 0.99]; aOR 0.69 [0.59 to 0.80]; 0.69 [0.60 to 0.81] for IMD quintile 2, 3 and 4 respectively compared to IMD quintile 1), and previously tested for HIV and/or syphilis (aOR 0.76 [0.68 to 0.86])

## Discussion

We provide data on the user characteristics of those undergoing online based STI testing and their subsequent rate of clinic attendance within a large patient cohort followed over 12-month study period. More online STI test users were asymptomatic (79%) compared to the clinic STI test users (49%). This is consistent with other studies where 79%-93% of online STI test service users were asymptomatic compared to 50%-69% of clinic STI test users [3, 4, 16, 17], probably reflecting guidance advising those with symptoms to attend clinic for a full evaluation or patient preference for a clinic appointment when symptoms are present. We explored the characteristics of STI test users and observed that regardless of the symptom status at the time of testing, specific STI risk groups such as MSM and those self-identifying as being of 'Black ethnicity' were significantly less likely to use the online STI test service, findings that are similar to some but not all previous studies [6, 16]. Concerns over the privacy of postal delivered STI self-sampling kits to homes where test users live with family or other people, and the perceived poorer user experience compared with the traditional face-to-face clinic testing may be of particular importance within ethnic and other minority groups [19]. Barnard et al [6] reported an increased uptake of online gonorrhoea and/or chlamydia testing in those self-identifying as MSM in some London boroughs, and it is possible that online test uptake may vary according to the specific characteristics of the online testing package or that geographic differences could exist. We found reduced utilisation of the online STI testing service amongst asymptomatic STI test users from the most socioeconomically deprived neighbourhoods. Evidence suggests that this may be linked to an increased risk of digital exclusion (reduced access to suitable IT hardware, costs of online access, limited digital skills [20]) or limited health literacy and lack of engagement [21]. There was, however, some inconsistency across socioeconomic categories with reduced use of online self-sampling also seen in the quintile with least deprivation, possibly reflecting the small proportions of these groups in our study and/or the lower STI risk associated with living in less deprived areas [22, 23]. Online based STI testing amongst symptomatic individuals was higher for those who were contacts of an STI, probably reflecting the local pathway used to expedite partner testing and treatment.

**Table 3. Factors associated with booking a sexual health clinic appointment within 90 days of an asymptomatic online STI test (n = 7769).**

| | | Booked clinic appointment within 90 days of online STI test | | Clinic appointment booked | | |
|---|---|---|---|---|---|---|
| | | No (n = 7285/7769 [93.8%]) | Yes (n = 484/7769 [6.2%]) | OR (95%CI) | aOR (95%CI) | P value |
| Age (median[IQR]), in years Min;Max | | 24(21 to 29) 16;75 | 25 (21 to 30) 16;67 | NA | NA | NA |
| Age, n(%) | 16 to 20 | 1380(18.9) | 103 (21.3) | 1 | 1 | **0.010** |
| | 21 to 30 | 4434 (60.9) | 270 (55.8) | 0.81 (0.62 to 1.04) | **0.75 (0.58 to 0.97)** | |
| | 31 to 40 | 1016 (14.0) | 78 (16.0) | 0.96 (0.68 to 1.35) | 0.87 (0.61 to 1.23) | |
| | Over 40s | 455 (6.3) | 33 (6.8) | 1.14 (0.75 to 1.75) | 0.64 (0.40 to 1.02) | |
| Gender/ sexuality, n(%)‡ | Female | 5033 (69.2) | 308 (63.6) | 1 | 1 | <**0.001** |
| | Male | 1576 (21.7) | 67 (13.8) | 0.71 (0.53 to 0.95) | 0.80 (0.59 to 1.08) | |
| | MSM | 667 (9.2) | 109 (22.5) | 2.67 (2.07 to 3.45) | **2.55 (1.58 to 4.09)** | |
| Ethnicity, n(%) | Asian | 338 (4.6) | 25 (5.2) | 1.24 (0.79 to 1.97) | 1.01 (0.63 to 1.61) | 0.657 |
| | Black | 556 (7.6) | 58 (12.0) | 1.57 (1.12 to 2.20) | 1.23 (0.87 to 1.73) | |
| | Not stated | 123 (1.7) | 5 (1.0) | 0.89 (0.38 to 2.08) | 0.70 (0.28 to 1.79) | |
| | Other | 714 (9.8) | 57 (11.8) | 1.33 (0.97 to 1.84) | 1.13 (0.81 to 1.57) | |
| | White | 5554 (76.2) | 339 (70.0) | 1 | 1 | |
| Index of Multiple Deprivation (IMD) Quintile, n(%)* | 20% (most deprived) | 2714 (37.4) | 214 (44.5) | 1 | 1 | **0.033** |
| | 20% to 40% | 2207 (30.4) | 159 (33.1) | 0.90 (0.71 to 1.14) | **0.77 (0.60 to 0.98)** | |
| | 40% to 60% | 1253 (17.2) | 61 (12.7) | 0.60 (0.43 to 0.83) | **0.69 (0.50 to 0.95)** | |
| | 60% to 80% | 652 (9.0) | 25 (5.2) | 0.46 (0.29 to 0.74) | **0.58 (0.36 to 0.91)** | |
| | 80 to 100% (least deprived) | 437 (6.0) | 22 (4.6) | 0.63 (0.39 to 1.04) | 0.85 (0.54 to 1.42) | |
| Contact of STI, n(%)* | No | 6664 (91.5) | 425 (88.2) | 1 | 1 | 0.759 |
| | Yes | 619 (8.5) | 57 (11.8) | 1.28 (0.93 to 1.75) | 0.95 (0.69 to 1.31) | |
| Method of STI testing in the last three months, n(%) | Clinic | 389 (5.3) | 152 (31.4) | 8.83 (6.97 to 11.2) | **5.65 (4.30 to 7.43)** | <**0.001** |
| | Kit | 839 (11.5) | 109 (22.5) | 3.5 (2.74 to 4.48) | **3.12 (2.42 to 4.03)** | |
| | No | 6057 (83.1) | 223 (46.1) | 1 | 1 | |
| Delivery method of online self-sampling STI test Kit, n(%) | Clinic | 605 (8.3) | 115 (23.8) | 3.53 (2.75 to 4.54) | **1.98 (1.46 to 2.68)** | <**0.001** |
| | Pharmacy | 429 (5.9) | 37 (7.6) | 1.70 (1.17 to 2.47) | **1.50 (1.03 to 2.20)** | |
| | Other(delivery partner or post) | 513 (7.0) | 19 (3.9) | 0.84 (0.52 to 1.34) | 0.79 (0.48 to 1.30) | |
| | Home | 5738 (78.8) | 313 (64.7) | 1 | 1 | |
| Tested for gonorrhoea/chlamydia, n(%) | No | 0(0) | 0(0) | NA | NA | NA |
| | Yes | 7285 (100) | 484 (100) | NA | NA | |

(*Continued*)

**Table 3.** (Continued)

| | | Booked clinic appointment within 90 days of online STI test | | Clinic appointment booked | | |
|---|---|---|---|---|---|---|
| | | No (n = 7285/7769 [93.8%]) | Yes (n = 484/7769 [6.2%]) | OR (95%CI) | aOR (95%CI) | P value |
| Tested for HIV/syphilis, n(%) | No | 2188 (30.0) | 204 (42.1) | 1 | 1 | **0.008** |
| | Yes | 5097 (70) | 280 (57.9) | 0.61 (0.50 to 0.75) | **0.74 (0.59 to 0.92)** | |
| Tested for hepatitis B, n(%) | No | 6705 (92.0) | 409 (84.5) | 1 | 1 | 0.070 |
| | Yes | 580 (8.0) | 75 (15.5) | 1.97 (1.47 to 2.63) | 0.60 (0.35 to 1.04) | |

aOR- adjusted for age, gender/sexuality, ethnicity, IMD Quintile, contact of STI, method of testing in the last three months, delivery method of self-sample test kit, HIV/syphilis test, hepatitis B test and an interaction term for gender/sexuality and hepatitis B. n = 7728/7769 observations.

‡ n = 9 non-binary gender were exclude due to the small proportion of this group.

*—Does not add up to column total due to missing records.

NA-Not assessed

IQR- interquartile range; Min- minimum; Max- maximum

Bold fonts highlight statistically significant associations (P<0.05)

Nearly half (46%) of the online self-sampling test kits in our cohort were not returned for processing which is consistent with previous studies where the non-return rates vary from 14% [24] to 63% [25]. Low return rates could miss infections and increase the overall cost per diagnosis [11]. Certain behavioural and demographic characteristics are associated with test kit return [6, 15, 17] which are also likely to reflect difficulties in sample self-collection (especially blood), lack of clarity around the sampling procedure, and reduced confidence in the testing process [15, 16]. Recent guidance for the design of self-sampling kits in England has made recommendations on redesigning kits to improve patient usability, and to increase clarity and confidence by providing patients with additional information on the online testing procedure [26]. In addition, optimised messaging and easy access to support may help to increase test kit return [27] although further studies are needed to confirm potential benefits.

Online STI test users who test positive for an STI are referred to a clinic for confirmatory tests (for HIV or syphilis) and/or treatment [6, 12, 15, 17]. However, anecdotal evidence suggests that some asymptomatic online test takers attend clinic, despite receiving a negative test result, for additional reassurance and further advice. We assessed the frequency of clinic attendance following an asymptomatic online STI test and found that 6.2% (484/7769) of online STI test users who were asymptomatic at the time of STI testing and who tested negative, subsequently booked clinic appointments within 90 days of their test with almost half (45% [220/484]) of these appointments made within 30 days of testing. In comparison, the frequency of re-attendance in the corresponding asymptomatic patients who initially attended the clinic was more than five times higher (33% [4960/15238]) which most likely reflects a more complex initial clinical presentation and the need for follow-up. This suggests that the risk of 'double dipping' (utilising both online and face to face clinic services within a short time period) is small, especially as the factors associated with subsequent clinic use suggest that clinic re-attendance may often be appropriate, for example, MSM attending for HIV pre-exposure prophylaxis. The fact that these groups of patients reuse clinic services soon after online STI testing may also reflect the difficulties in self-sampling blood and can explain the lower uptake of online HIV/syphilis testing in this study, and further affect re-attendance rates particularly if individuals lack confidence in having provided adequate samples for testing. However, it is also possible that subsequent clinic attendance could be associated with unmet sexual health

**Table 4. Factors associated with sexual clinic booking appointment within 90 days of an asymptomatic clinic STI test (n = 15238).**

| | | Booked clinic appointment within 90 days of a clinic STI test | | Clinic appointment booked | | |
|---|---|---|---|---|---|---|
| | | No (n = 10278/15238 [67.4%]) | Yes (n = 4960/15238 [32.6%]) | OR(95%CI) | aOR(95%CI) | P value |
| Age in years, median (IQR) Min;Max | | 26(21 to 32) 16;81 | 26 (21 to 33) 16;74 | NA | NA | NA |
| Age, n(%) | 16 to 20 | 2017 (19.6) | 1072 (21.6) | 1 | 1 | <0.001 |
| | 21 to 30 | 5199 (50.6) | 2379 (48.0) | 0.85 (0.78 to 0.93) | **0.80 (0.73 to 0.88)** | |
| | 31 to 40 | 1963 (19.1) | 952 (19.2) | 0.92 (0.83 to 1.03) | **0.80 (0.71 to 0.90)** | |
| | Over 40s | 1099 (10.7) | 557 (11.2) | 0.96 (0.85 to 1.10) | **0.82 (0.71 to 0.94)** | |
| Gender/sexuality, n(%)‡ | Female | 5250 (51.1) | 2274 (45.9) | 1 | 1 | <0.001 |
| | Male | 4292 (41.8) | 1680 (34.0) | 0.92 (0.85 to 1.00) | **0.89 (0.81 to 0.98)** | |
| | MSM | 724 (7.1) | 992 (20.1) | 3.28 (2.93 to 3.67) | **3.41 (2.97 to 3.92)** | |
| Ethnicity, n(%) | Asian | 1033 (10.1) | 578 (11.7) | 1.19 (1.06 to 1.34) | **1.19 (1.05 to 1.35)** | <0.001 |
| | Black | 2149 (20.9) | 1088 (21.9) | 1.06 (0.97 to 1.17) | **1.15 (1.03 to 1.27)** | |
| | Not stated | 1759 (17.1) | 740 (14.9) | 0.89 (0.80 to 0.98) | 0.90 (0.81 to 1.01) | |
| | Other | 1039 (10.1) | 532 (10.7) | 1.10 (0.97 to 1.24) | 1.11 (0.97 to 1.26) | |
| | White | 4298 (41.8) | 2022 (40.8) | 1 | 1 | |
| Index of Multiple Deprivation (IMD) Quintile, n(%)* | 20% (most deprived) | 4779 (47.3) | 2525 (51.6) | 1 | 1 | <0.001 |
| | 20% to 40% | 2325 (23.0) | 1130 (23.1) | 0.93(0.85 to 1.02) | **0.89 (0.82 to 0.98)** | |
| | 40% to 60% | 1314 (13.0) | 630 (12.9) | 0.91 (0.81 to 1.01) | **0.88 (0.78 to 0.99)** | |
| | 60% to 80% | 832 (8.3) | 309 (6.3) | 0.70 (0.60 to 0.80) | **0.69 (0.59 to 0.80)** | |
| | 80 to 100% (least deprived) | 853 (8.4) | 302 (6.2) | 0.67 (0.58 to 0.78) | **0.69 (0.60 to 0.81)** | |
| Contact of STI, n(%)* | No | 8892 (86.5) | 4256 (85.8) | 1 | 1 | 0.367 |
| | Yes | 1089 (10.6) | 558 (11.3) | 1.07 (0.96 to 1.20) | 0.92 (0.82 to 1.04) | |
| | Unknown | 297 (2.9) | 146 (2.9) | 0.99 (0.81 to 1.21) | 1.04 (0.84 to 1.29) | |
| Attended clinic in the last three month (at baseline), n(%)* | No | 4430 (43.1) | 1953 (39.4) | 1 | 1 | **0.015** |
| | Yes | 5848(56.9) | 3007 (60.6) | 1.08 (1.01 to 1.15) | **1.10 (1.02 to 1.18)** | |
| Tested for gonorrhoea/chlamydia, n(%) | No | 327 (3.2) | 197 (4.0) | 1 | 1 | 0.186 |
| | Yes | 9951 (96.8) | 4763 (96.0) | 0.82 (0.69 to 0.99) | 0.88 (0.73 to 1.07) | |
| Tested for HIV /syphilis, n(%) | No | 1080 (10.5) | 594 (12.0) | 1 | 1 | <0.001 |
| | Yes | 9198 (89.5) | 4366 (88.0) | 0.89 (0.80 to 0.99) | **0.76 (0.68 to 0.86)** | |

(*Continued*)

**Table 4.** (Continued)

| | | Booked clinic appointment within 90 days of a clinic STI test | | Clinic appointment booked | | |
|---|---|---|---|---|---|---|
| | | No (n = 10278/15238 [67.4%]) | Yes (n = 4960/15238 [32.6%]) | OR(95%CI) | aOR(95%CI) | P value |
| Tested for Hepatitis B, n(%) | No | 7710 (75.0) | 3372 (68.0) | 1 | 1 | <0.001 |
| | Yes | 2568 (25.0) | 1588 (32.0) | 1.46 (1.36 to 1.58) | **1.41(1.25 to 1.58)** | |

aOR- adjusted for age, gender/sexuality, ethnicity, IMD Quintile, contact of STI, clinic attendance in the last three months, gonorrhoea/chlamydia test, HIV/syphilis test, hepatitis B test and an interaction term for gender/sexuality and hepatitis, n = 14975/15238 observations

‡ n = 26 non-binary gender were exclude due to being a small proportion of this group

*—Does not add up to column total due to missing records

NA-Not assessed

Bold fonts highlight statistically significant associations (P<0.05)

needs including sexual assault or need for specialist contraception advice [28, 29]. The economic case for online testing is supported by a low rate of re-attendance and the associated costs at an individual patient level [11]. However, the provision and increased accessibility of online testing may lead to an increase in overall service demand and activity which could increase service costs overall.

Our study has some potential limitations. The dataset is limited to attendees at a single large sexual health service which may limit its generalisability. However, Umbrella is one of the largest providers of online testing services in the UK and covers a diverse area with respect to individual demographics and sexual behaviours. The clinic STI testing cohort was limited to sexual health clinic attendees and does not include testing that occurred in general practices or other healthcare settings. This study was a retrospective analysis of routinely collected data, therefore the findings may be subject to unidentified selection bias including the exclusion of 3758 individuals in whom a specific STI test could not be linked to their clinic attendance. Data over a single one year time period was analysed resulting in an unequal follow up time period for included patients although the median time to re-attendance of around 30 days suggests that our findings remain valid. The data reported predates the COVID 19 pandemic and the impact of related changes in sexual healthcare delivery remains unclear. Finally, multiple statistical testing was performed without adjustment for multiplicity, which raises the possibility of spurious results and cautious interpretation is required.

In summary, we found that uptake of online testing services was reduced amongst individuals who were the most socio-economically deprived, in addition to MSM and ethnic minorities. However, subsequent clinic attendance following a negative online test rest was relatively uncommon (6%) suggesting that patient needs were being met without placing a large additional burden on clinic based services. As remote consultations and digital healthcare become more common, a greater understanding of the barriers and facilitators for online based testing are required to inform the development of optimised care pathways and specific interventions to ensure equality of access.

## Supporting information

**S1 Checklist. STROBE statement—checklist of items that should be included in reports of observational studies.**
(DOCX)

## Acknowledgments

The authors thank James Pye, Aisha Diler for retrieving the study data from the electronic patient record and internet testing website database, and James Hodson for support with the statistical analysis.

## Author Contributions

**Conceptualization:** Oluseyi Ayinde, Louise Jackson, Jonathan D. C. Ross.

**Formal analysis:** Oluseyi Ayinde.

**Methodology:** Oluseyi Ayinde, Louise Jackson, Jara Phattey, Jonathan D. C. Ross.

**Writing – original draft:** Oluseyi Ayinde.

**Writing – review & editing:** Oluseyi Ayinde, Louise Jackson, Jara Phattey, Jonathan D. C. Ross.

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
