## [Decision Letter · Decision Letter 0]

1 Sep 2022

PONE-D-22-04602STI testing and subsequent clinic attendance amongst test negative asymptomatic users of an internet STI testing service; one-year retrospective studyPLOS ONE

Dear Dr. Ayinde,

Thank you for submitting your manuscript to PLOS ONE. After careful consideration, we feel that it has merit but does not fully meet PLOS ONE’s publication criteria as it currently stands. Therefore, we invite you to submit a revised version of the manuscript that addresses the points raised during the review process.

Please note that we have only been able to secure a single reviewer to assess your manuscript. We are issuing a decision on your manuscript at this point to prevent further delays in the evaluation of your manuscript. Please be aware that the editor who handles your revised manuscript might find it necessary to invite additional reviewers to assess this work once the revised manuscript is submitted. However, we will aim to proceed on the basis of this single review if possible. The reviewer has raised a particular concern regarding the follow-up period used in your study, and has asked that you explain your rationale for this aspect of the study design. Please respond carefully to this request, as well as the other points the reviewer has raised.

We look forward to receiving your revised manuscript.

Kind regards,

Jamie Males

Editorial Office

PLOS ONE

Journal Requirements:

a) Did participants provide their written or verbal informed consent to participate in this study?

“JDCR reports personal fees from Bayer and GSK Pharma, as well as ownership of shares in GSK Pharma and AstraZeneca Pharma; and is author of the UK and European Guidelines on Pelvic Inflammatory Disease; is a Member of the European Sexually Transmitted Infections Guidelines Editorial Board. He is an NIHR Journals Library Editor and associate editor of Sexually Transmitted Infections journal. He is an officer of the International Union against Sexually Transmitted Infections (treasurer), and a charity trustee of the Sexually Transmitted Infections Research Foundation (chair). OA, LJ, and JP declare no competing interests or funding.”

We note that one or more of the authors are employed by a commercial company: name of commercial company.

Reviewers' comments:

Reviewer's Responses to Questions

**Comments to the Author**

1. Is the manuscript technically sound, and do the data support the conclusions?

Reviewer #1: Yes

2. Has the statistical analysis been performed appropriately and rigorously? 

Reviewer #1: Yes

3. Have the authors made all data underlying the findings in their manuscript fully available?

Reviewer #1: No

4. Is the manuscript presented in an intelligible fashion and written in standard English?

Reviewer #1: Yes

5. Review Comments to the Author

Reviewer #1: This manuscript outlines a retrospective study of users of Internet STI testing service and sexual health clinics. The study found that there was a low subsequent clinic attendance amongst online STI test service users compared to STI clinics user.

I think the paper is well written, the methods are sound and appropriate conclusion points are drawn from the data used. However, my main concern is the follow up period used. By using one calendar year of data there is unequal follow up time for testers which is a significant limitation and is not mentioned in the limitations. Additionally, at the time of the study I believe only gay, bisexual and other men who have sex with men were recommended to test 3 monthly, all sexually active men and women aged 15-24 should be tested for chlamydia annually or on change of sexual partner (whichever is more frequent). Therefore, the people who reattend within 90 days would be a select group of people who may be at higher STI risk and the results might not be generalisable to other populations. I would suggest the explain the rationale for using a 90 day cut offs or use a longer time period for follow up in order to be representative of recommended testing guidelines.

Additionally, the references to the proportion of asymptomatic users of online testing services need to be clearly caveated that this metric will be biased because most people who are symptomatic are asked to go to a clinic and not use self-sampling. Especially as some services are only commissioned for asymptomatic people only.

Specific comments

• In some parts you mention the risk of subsequent attendance. Odds ratios do not indicate the ‘risk’ but the likelihood of something occurring.

• There are several places where acronyms are not spelt out

• Tables – there are some formatting issues; missing (95%CI), not sure why some cells are in bold.

• Limitations - have not mentioned that you analysis will not include people attending other sexual health services.

• Point to comment on, lines 260-1: “those self-identifying as being of ‘Black ethnicity’ were significantly less likely to use the online STI test service.” Some studies found that self-sampling was difficult for some communities/ethnic groups where the test being delivered to a home address with other family members who lack of privacy. Could use this references to explain this. For example the link below https://bmcpublichealth.biomedcentral.com/articles/10.1186/s12889-018-5256-5

6. PLOS authors have the option to publish the peer review history of their article (what does this mean?). If published, this will include your full peer review and any attached files.

Reviewer #1: No

---

## [Author Response · Author response to Decision Letter 0]

15 Sep 2022

We thank the Editor and Reviewer for their comments and have revised the manuscript to address their comments. A rebuttal 'Response to reviewer' is enclosed with this submission.

---

## [Decision Letter · Decision Letter 1]

23 Jan 2023

STI testing and subsequent clinic attendance amongst test negative asymptomatic users of an internet STI testing service; one-year retrospective study

PONE-D-22-04602R1

Dear Dr. Ayinde,

We’re pleased to inform you that your manuscript has been judged scientifically suitable for publication and will be formally accepted for publication once it meets all outstanding technical requirements.

Kind regards,

Habakkuk Yumo, MD, MSc, PhD

Academic Editor

PLOS ONE

Additional Editor Comments (optional):

Reviewers' comments:

Reviewer's Responses to Questions

**Comments to the Author**

1. If the authors have adequately addressed your comments raised in a previous round of review and you feel that this manuscript is now acceptable for publication, you may indicate that here to bypass the “Comments to the Author” section, enter your conflict of interest statement in the “Confidential to Editor” section, and submit your "Accept" recommendation.

Reviewer #1: All comments have been addressed

Reviewer #2: All comments have been addressed

2. Is the manuscript technically sound, and do the data support the conclusions?

Reviewer #1: Yes

Reviewer #2: Yes

3. Has the statistical analysis been performed appropriately and rigorously? 

Reviewer #1: Yes

Reviewer #2: Yes

4. Have the authors made all data underlying the findings in their manuscript fully available?

Reviewer #1: Yes

Reviewer #2: No

5. Is the manuscript presented in an intelligible fashion and written in standard English?

Reviewer #1: Yes

Reviewer #2: Yes

6. Review Comments to the Author

Reviewer #1: Thank you for addressing my previous main concern and specific comments. I am satisfied with your added explanation in the text and I think it sets out the rationale for the study in a much clearer way.

I have some further specific (minor) comments to add which I think can easier be fixed

Title

• Now you have updated the rationale for the study, you are not following people up for a year so I would say it is a retrospective cohort study

• I would also make note of the comparison aspect to the study (online testers vs clinic testers)

Abstract

• Lines 31 and 32 – compared to who?

• Lines 36 – this metric is a bit biased as you had to have previously tested to be included in the study

Table 1

• I think it will be worth rephrasing ‘Gender/Sexuality’ and updating the footnotes as they aren’t the most clear.

o Gender - It would also be useful to confirm if it is gender and not sex

o Sexuality – MSM is not a sexuality but a sexual behaviour

o The footnote for Men does not add up to the total for Male and Bisexual men appear to be included in the Male breakdown.

o My suggested change would be ‘Sex’ and Sexual orientation with All women, Heterosexual men and gay, Bisexual and MSM.

• IMD breakdown – categories are overlapping, i.e. 20% in first and second group, 40% in second and third

• Clinic attendance in the last three months – I’ve mentioned this in the abstract section but it seems a bit circular. Would this include people who tested in clinic and then tested online? Please clarify the definition further

Discussion

• Line 227 Add ‘certain’ before ‘ethnic and..’

Limitation

• People might have tested online too early in the window period of their infection and so would have been too early to tell.

Reviewer #2: Thanks for the clarifications with respect to the review comments and for the changes made in the manuscript.

7. PLOS authors have the option to publish the peer review history of their article (what does this mean?). If published, this will include your full peer review and any attached files.

Reviewer #1: **Yes: **Sophie Grace Nash

Reviewer #2: No

---

## [Editor Report · Acceptance letter]

30 Jan 2023

PONE-D-22-04602R1 

STI testing and subsequent clinic attendance amongst test negative asymptomatic users of an internet STI testing service; one-year retrospective study 

Dear Dr. Ayinde:

I'm pleased to inform you that your manuscript has been deemed suitable for publication in PLOS ONE. Congratulations! Your manuscript is now with our production department. 

Kind regards, 

on behalf of

Dr. Habakkuk Yumo 

Academic Editor

PLOS ONE